# Anxiolytic and Antidepressant Effects of the Hydroethanolic Extract from the Leaves of *Aloysia polystachya* (Griseb.) Moldenke: A Study on Zebrafish (*Danio rerio*)

**DOI:** 10.3390/ph12030106

**Published:** 2019-07-11

**Authors:** Nayara Costa de Melo, Brenda Lorena Sánchez-Ortiz, Tafnis Ingret dos Santos Sampaio, Arlindo César Matias Pereira, Fernando Luiz Pinheiro da Silva Neto, Heitor Ribeiro da Silva, Rodrigo Alves Soares Cruz, Hady Keita, Ana Maria Soares Pereira, José Carlos Tavares Carvalho

**Affiliations:** 1Laboratório de Pesquisa em Fármacos, Curso de Farmácia, Departamento de Ciências Biológicas e da Saúde, Universidade Federal do Amapá, Macapá, Amapá, CEP 68.903-419, Brasil; 2Programa de Pós-Graduação em Inovação Farmacêutica, Departamento de Ciências Biológicas e da Saúde, Universidade Federal do Amapá, Macapá, Amapá, CEP 68.903-419, Brasil; 3Departamento de Farmacia, Facultad de Química, Universidad Nacional Autónoma de México, Ciudad Universitaria, Ciudad de México, C.P. 04510, México; 4Programa de Pós-Graduação em Ciências Farmacêuticas, Departamento de Ciências Biológicas e da Saúde, Universidade Federal do Amapá, Macapá, Amapá, CEP 68.903-419, Brasil; 5Universidad de la Sierra Sur, Division de Pós-Grado, Instituto de Investigación Sobre la Salud Pública, Ciudad Universitaria, Oaxaca, C.P. 70800, México; 6Departamento de Biotecnologia em Plantas Medicinais, Universidade de Ribeirão Preto (UNAERP), Ribeirão Preto, São Paulo, CEP 14096-900, Brasil; 7Rede Bionorte, Programa de Pós-Graduação em Biotecnologia, Universidade Federal do Amapá, Macapá, Amapá, CEP 68.903-419, Brasil

**Keywords:** *Aloysia polystachya*, anxiolytic, antidepressant, *Danio rerio*

## Abstract

Medicinal plants such as *Aloysia polystachya* are often used in the treatment of psychiatric diseases, including anxiety- and depression-related humor disturbances. In folk medicine, *A. polystachya* is used to treat digestive and respiratory tract disturbances, as a sedative and antidepressant agent, and as a tonic for the nerves. This study aimed to evaluate the antidepressant and anxiolytic effect from the hydroethanolic extract from the leaves of *Aloysia polystachya* (HELAp) in zebrafish. The extract was analyzed through ultra-performance liquid chromatography-mass spectroscopy (UPLC-MS) and the main compound detected was acteoside. HELAp was administered orally (10 mg/kg) and through immersion (mg/L). The anxiolytic activity was evaluated through the scototaxis (light–dark) test using caffeine as an anxiogenic agent and buspirone as a positive control. The parameters assessed were: period spent in the white compartment (s), latency (s), alternations (n), erratic swims (n), period of freezing (s), thigmotaxis (s), and risk evaluation (n). The antidepressant effect was evaluated through the novel tank diving test using 1% ethanol, unpredictable chronic stress, and social isolation as depressors; fluoxetine was used as a positive control. The parameters assessed were: period spent at the top of the tank, latency, quadrants crossed, erratic swim, period of freezing, and distance of swam. The main chemical compound of HELAp was acteoside. The administration of the extract on zebrafish managed to revert the anxiogenic effect of caffeine without impairing their locomotion. Additionally, the treatment exerted antidepressant activity similarly to fluoxetine. Overall, the results suggest a significant anxiolytic and antidepressant activity to the extract, which is probably due to the presence of the major compound, acteoside.

## 1. Introduction

Mental and behavioral disorders are currently among the major causes of disability due to disturbances of affected people’s mood and feelings. Nowadays, there is a growing prevalence of anxiety and depression, which could ultimately lead to suicide. According to the World Health Organization (WHO), 3.6% of people in the world suffer from anxiety disorder (about 264 million people), and 4.4% suffer from depression (about 322 million people); in 2020, depression will be the leading disabling disease on the world [1]. Different stress situations occur during the lifetime, and these situations can affect individuals differently, resulting in non-specific psychopathological manifestations such as depression and anxiety [2,3]. In fact, chronic stress is an animal model of depression [4].

An essential approach in drug development is the research of bioactive molecules derived from plants to develop alternative treatments and develop new drugs, which could be useful, including for psychiatric treatments. There are several plants used in folk medicine used to relieve anxiety and depression, for instance, *Aloysia polystachya* (Griseb.) Moldenke (Verbenaceae), a plant found in South America, popularly known as “burrito”. The leaves of this plant have some known properties, such as digestive activity [5], sedative and tranquilizer [6,7], and antidepressant activity in mice [8] and rats [9]. 

The most representative compounds of this species are phenylethanoid and phenylpropanoid glycosides such as forsythoside A, plantainoside C, martynoside [10], and, mainly, the acteoside [10,11]. Phenylpropanoids are water-soluble polyphenols, largely found in medicinal plants. Plants from the genus Aloysia, including *A. polystachya*, are a rich source of phenylpropanoid heterosides containing a disaccharide bound to a phenylethanoid and phenylpropanoid moiety [10,12,13]. The acteoside, also known as verbascoside, is a water-soluble phenylpropanoid glycoside with reducing and radical scavenging ability in a hydrophobic medium (such as the membranes), acting in cell signaling and mitochondrial function [14,15,16]. Recently, it has been reported that acteoside, a molecule found in the extract of *Aloysia polystachya*, has inhibitory activity over the monoamine oxidase A [11].

Zebrafish (*Danio rerio*) has been used as a valuable model to perform toxicological studies and behavioral analysis of stress, fear, anxiety, depression, among others. The neurochemistry of zebrafish makes it a useful model to test drugs with anxiolytic or anxiogenic activity due to its high homology with mammals [17,18,19,20,21,22,23,24,25,26,27,28,29]. The digestive tract of zebrafish includes mouth, pharynx, esophagus, intestine, and anal opening, without a stomach (indicating that acidification is not necessary for this species); it is believed that the intestinal bulb substitutes the stomach role. The intestine is located in the abdominal cavity, and it is divided into three parts: anterior, middle, and posterior. The anterior intestine (also known as the intestinal bulb) has the function of absorption and food storage [30,31,32]. In the mucosal layer is found goblet cells, dispersed inflammatory cells, and enterocytes. Through these cells, the intestine manages to absorb nutrients (evidenced by the high content of digestive enzymes in this area and the height from the epithelial folds), act on the immune defense, and in the osmotic balance [33]. 

The gills are responsible for gas exchange in the species. Additionally, it has a role in acid–base balance, excretion of toxic residues, and ionic balance. In immersion treatments, this organ acts in the absorption of the compounds and as an indicator of contamination since it is in direct contact with the exterior medium; it is highly sensitive to unfavorable conditions [30,34,35].

Based on this information, this study aimed to evaluate the anxiolytic and antidepressant activity of the hydroethanolic extract from the leaves of *Aloysia polystachya* (HELAp) in zebrafish. For this, the extract was administered orally and through immersion, and the fishes were appraised in the scototaxis (light–dark) and novel tank diving tests.

## 2. Results

### 2.1. HELAp Analysis (UPLC-MS and HPLC-DAD)

The extract was analyzed through ultra-performance liquid chromatography-mass spectroscopy (UPLC-MS), and the chromatogram is shown in Figure 1. The chromatogram shows that the main compound is acteoside, and its concentration in the HELAp—determined through HPLC—was 99.74 ± 1.7 µg/mg of the dry extract. The ratio of peak area of standard acteoside to the corresponding concentration of analyte was established through linear regression of the standard curves. Limits of detection (LoD) and quantitation (LoQ) of acteoside were 0.27 and 0.89 μg/mL, respectively, (Figure 2).

### 2.2. Scototaxis Test (Light–Dark Test)

As for the time spent in the light compartment, the results show that animals treated with buspirone through immersion and orally had a high time in the light compartment (71.76% ± 5.51% and 79.91% ± 5.26%, respectively), spending more time in this compartment; the group treated with HELAp also spent more time in the light compartment (57.27% ± 5.35% and 65.39% ± 5.36%, respectively). On the other hand, the group treated only with caffeine spent significantly less time (11.83% ± 2.71% and 10.32% ± 2.77%) in the light compartment compared to the control group (*p* < 0.01, Figure 3).

As for the number of alternations, the negative control groups had average numbers (7.75 ± 0.96 and 10.75 ± 0.96), while the groups treated only with caffeine had significantly lower numbers (4.25 ± 0.96 and 5.75 ± 0.58). In contrast, the groups treated with buspirone through immersion and orally had a higher number of alternations (17 ± 0.81 and 18.5 ± 0.5), and so had the groups treated with HELAp (15.25 ± 0.5 and 17 ± 0.82; *p* < 0.01, Figure 4).

Animals treated with buspirone had lower latency period to enter the light compartment (149.05 ± 12.80 s and 120.13 ± 10.24 s), and the same was observed in the groups treated with the extract (157 ± 9.64 s and 141.15 ± 14.33 s). In contrast, the groups treated with caffeine had significantly higher latency period (344.33 ± 11.04 s and 318 ± 13.65 s; *p* < 0.01, Figure 5).

Regarding the period of freezing in the light compartment, the animals treated through immersion and orally with buspirone (19.95 ± 3.36 s and 18.98 ± 3.16 s), and HELAp (25.40 ± 4.17 s and 23.85 ± 2.03 s), had values similar to the negative control groups (20.45 ± 2.72 s and 17.25 ± 1.71 s), while the groups treated with caffeine had high values of freezing period (60.73 ± 4.21 s and 66.83 ± 3.21 s; *p* < 0.01, Figure 6).

In the assessment of number of erratic swims, the groups treated with buspirone through immersion and orally had the lowest values (4.50 ± 0.58 and 6 ± 0.82); the groups treated with HELAp also had low values (4 ± 0.82 and 3.25 ± 0.96), while the groups treated with caffeine had high values of this parameter (11 ± 1.15 and 12 ± 0.82; *p* < 0.01, Figure 7).

Compared to the control groups (34.70 ± 0.73 s and 33.13 ± 0.56 s), the period of thigmotaxis was significantly higher for animals treated with caffeine (132.45 ± 1.15 s and 130.03 ± 1.35 s), whilst the groups treated with buspirone (48 ± 0.80 s and 47 ± 0.79 s) and HELAp (52.60 ± 0.68 s and 50.95 ± 0.70 s) had lower values (*p* < 0.01, Figure 8).

In risk evaluation assessment, the groups treated with buspirone (6.75 ± 0.96 and 4.25 ± 0.96) and HELAp (8.50 ± 0.58 e 5.50 ± 0.58) had significantly lower values compared to the groups treated only with caffeine (18 ± 0.82 and 20 ± 0.82; *p* < 0.01, Figure 9).

### 2.3. Novel Tank Diving Test (NTDT)

In the novel tank test, the group treated with 1% ethanol spent 20.19% ± 2.67% (immersion) and 13.53% ± 2.67% (oral) of the time in the tank top, the animals subjected to stressors spent 22.67% ± 3.30% and 16.0%1 ± 3.30%, respectively, and the animals subjected to isolation spent 21.21% ± 2.79% and 17.87% ± 2.79%, respectively. This was reverted when animals were treated with fluoxetine in the groups treated with 1% ethanol (79.87% ± 3.54% and 83.20% ± 3.54%), stressors (83.84% ± 2.79% and 87.17% ± 2.79%), and social isolation (79.99% ± 2.89% and 80.22% ± 3.36%). The same was observed in animals treated with HELAp in the group treated with 1% ethanol (67.33% ± 3.82% and 67.08% ± 3.42%), stressors (75.43% ± 2.78% and 78.77% ± 2.78%), and social isolation (74.37% ± 4.25% in A and 76.03% ± 4.25% in B); all the groups spent >60% of the time in the tank top (*p* < 0.01, Figure 10).

There was a significant increase of latency period to reach the top in the groups treated with 1% ethanol (240.18 ± 8.72 s and 259.68 ± 11.06 s), in the groups subjected to stressors (226.24 ± 8.79 s and 214.74 ± 8.70 s), and in the socially-isolated animals (177.91 ± 9.67s and 180.57 ± 7.65 s); these groups had decreased period of time spent in the tank top. However, this could be reversed when fluoxetine was administered in the group treated with ethanol (18.78 ± 6.22 s and 8.43 ± 5.56 s), stressors (13.09 ± 6.24 s and 14.09 ± 5.06 s), and social isolation (12.95 ± 6.50 s and 19.95 ± 8.62 s). Accordingly, animals treated with the extract had decreased values of latency—ethanol: 26.48 ± 6.58 s and 12.68 ± 5.45 s; stressors: 13.09 ± 6.24 s and 14.09 ± 5.06 s; social isolation: 20.53 ± 6.70 s and 19.03 ± 8.48 s (*p* < 0.01, Figure 11).

As for the number of quadrants crossed, the groups treated with 1% ethanol (26.75 ± 2.50 and 18.25 ± 2.59), stressors (33.50 ± 3.42 and 32.25 ± 3.96), and social isolation (45.75 ± 2.63 and 36.5 ± 3.84) had fewer values, while those co-treated with fluoxetine had higher values—1% ethanol: 140.50 ± 3.42 and 143.25 ± 4.15; stressors: 124.50 ± 4.43 and 136.5 ± 3.35; social isolation: 114 ± 4.16 and 124.25 ± 4.32. The groups co-treated with HELAp also had higher values of quadrant crossings—1% ethanol: 124 ± 4.08 and 123.75 ± 3.96; stressors: 115 ± 3.37 and 112.25 ± 3.70; social isolation 93.75 ± 4.57 and 109.5 ± 3.84 (*p* < 0.01, Figure 12).

Significant differences were also observed regarding the number of erratic swims. The groups treated with 1% ethanol (28.5 ± 1.29 and 28.5 ± 1.29), stressors (25.50 ± 1.29 and 21.5 ± 1.29), and social isolation (21.07 ± 0.83 and 25.5 ± 1.29) had high values of this parameter, while the group co-treated with fluoxetine had lower values—1% ethanol: 140.50 ± 3.42 and 143.25 ± 4.15; stressors: 124.50 ± 4.43 and 136.5 ± 3.35; social isolation 114 ± 4.16 e 124.25 ± 4.32. Co-treatment with HELAp also resulted in decreased number of erratic swims—1% ethanol: 4.75 ± 0.96 and 4.5 ± 1.29; stressors: 5.75 ± 0.96 and 5.5 ± 1.29; social isolation: 8 ± 0.82 and 6.75 ± 0.96 (*p* < 0.01, Figure 13).

The periods of freezing in the groups without antidepressant treatment were high: 29.50 ± 1.29 s and 35.75 ± 0.96 s for 1% ethanol; 26.75 ± 0.96 s and 20.31 ± 1.22 s for stressors; and 23.10 ± 1.17 s and 23.21 ± 1.38 s for social isolation. The groups co-treated with fluoxetine had lower periods of freezing: 6.75 ± 0.96 s and 5 ± 0.82 s for 1% ethanol; 8.19 ± 0.87 s and 6.55 ± 1.05 s for stressors; 9.80 ± 1.11 s and 8.22 ± 0.91 s for social isolation. The same occurred with groups treated with the HELAp—8.25 ± 0.96 s and 7 ± 0.82 s for 1% ethanol; 11.34 ± 0.79 s and 8.87 ± 1.07 s for stressors; 11.95 ± 1.37 s and 9.61 ± 1.02 s for social isolation (*p* < 0.01, Figure 14).

Lastly, the groups treated with 1% ethanol, stressors, and social isolation had significantly lower values of swam distance, evidencing decreased locomotor activity, while the groups co-treated with fluoxetine and HELAp had significantly higher values (*p* < 0.01, Figure 15).

## 3. Discussion

HELAp has acteoside as its primary compound; this molecule has some known biological activities such as antioxidant and free radical scavenging [36,37,38,39,40,41,42,43,44,45,46,47], antiproliferative [48], cytoprotective [15,49], and anti-inflammatory [50,51]. Moreover, acteoside has a neuroprotective effect against 1-methyl-4-phenyl-pyridinium (MPP)-induced and mitochondrial disfunction-induced apoptosis. Several mechanisms can be involved in such protection, and its antioxidant activity is probably involved [36].

The anxiolytic evaluation in fish through light–dark preference tests was first described by Serra et al. [52]. This method is based in the higher period of exploration in the dark side of the apparatus, in the higher period of latency to reach the white compartment when the animals are placed in the black compartment, and in the lower latency for animals to get from the white to the black compartment. Due to fishes’ natural aversion to illuminated environments, this model allows the researcher to evaluate the level of anxiety in the animals through the frequency of transitions and length of stay in the illuminated compartment.

Zebrafish’s stereotyped anxiogenic behavior, when placed on an aversive environment, is a well-known and accepted method to test novel anxiolytic drugs [3,4,53,54,55,56,57]. Hence, it corroborates the results of time spent in the white compartment.

Steenbergen et al. [26], testing buspirone in zebrafish through immersion (25 mg/L), reported that the treatment exerted anxiolytic activity in the fishes evidenced by the light–dark test. The same occurred in this study, where the treatments with buspirone and HELAp attenuated the caffeine-induced anxiogenic effect, evidenced by significant increase of time spent in the white compartment compared to the negative control groups treated only with water. In contrast, the groups treated only with caffeine caused a significant decrease in time spent in the white compartment (*p* < 0.01; Figure 3).

The effect of anxiolytic agents in this model, evidenced by increased time spent in the white compartment, is due to their motivation to explore new environments (scototaxis). On the other hand, their motivation in an anxiogenic state is to spend more time in the black (dark) compartment, as a way to seek protection instead of exploring. Anxiolytic agents such as buspirone manage to decrease the fishes’ anxiety, decreasing the period spent in the black compartment [53,56,57,58,59].

In the elevated plus maze test, Hellión-Ibarrola et al. [7] reported anxiolytic activity using a hydroethanolic extract from the leaves of *A. polystachya* in mice (orally) without sedative effects or impairing their locomotor activity. The treatment managed to increase the time of exploration and increase the period with open arms (an anxiolytic behavior). In accordance, Mora et al. [9] also reported anxiolytic effect caused by treatment with the hydroethanolic extract of *A. polystachya* in rats in the plus-maze test; however, in high doses, the treatment caused sedative activity in the open field test.

The changes in fishes’ preference of light or dark environments caused by caffeine (anxiogenic activity) are due to glutamatergic excitotoxicity caused by the blockage of A1 receptors [27,31,53,60,61,62]. As for the number of alternations, defined as the number of times that the fish’s pectoral fin crosses the central line [53], it was observed that groups treated only with caffeine had decreased values of this parameter, while the groups co-treated with an anxiolytic agent (such as HELAp or buspirone) had higher values. This is in accordance with Maximino et al. [60], Bencan et al. [63], Magno et al. [64], and Gebauer et al. [65]. In our study it was observed in the groups treated only with caffeine that the lower number of alternations (decreased mobility) was connected with the time spent in the dark compartment, which evidences an anxiogenic state [66,67,68] (*p* < 0.01; Figure 4). These behavioral changes of light–dark preferences also have been used to evaluate the toxicity of compounds [69].

Freezing is a condition characterized by the complete paralysis of the fish (except of operculum and eyes), indicating that the animal is in an aversive situation [70]; this parameter increased in the groups treated only with caffeine, evidencing increased anxiety, which is in accordance with Maximino et al. [59] and Kalueff et al. [71]. Co-treatment with HELAp significantly decreased the period of freezing, evidencing its anxiolytic activity (*p* < 0.01; Figure 6).

The erratic swim is a condition of multiple occurrences of high acceleration and changes of direction, apparently in a stochastic way [59,70,71]. Regarding this parameter, it was observed that the treatment with caffeine significantly increased its occurrence (*p* < 0.01; Figure 7), compared to the groups co-treated with buspirone or HELAp.

Thigmotaxis is defined as a swim near the walls (2 cm at most), and the avoidance of open areas by the fish [59]. This behavior occurs when the fish seek an opportunity of scape and is more frequent in the black compartment since the walls of the white compartment are visible [70]. Animals with a higher aversion to the white compartment also have higher values of thigmotaxis when placed in the black compartment [20,29,60,62]. In this study, the groups treated with buspirone or HELAp had significantly lower values of thigmotaxis compared to the group treated only with caffeine (*p* < 0.01; Figure 8)

Risk evaluation is defined as fast and quick entries in the white compartment (<1 s) followed by the return to the black compartment, or partial entry in the white compartment (when the pectoral fin does not cross the central line) [71]. The groups treated with buspirone or HELAp had lower values of risk evaluation (*p* < 0.01; Figure 9).

This study differs from previous studies performed in rats or mice [7,8,9], considering that eight different parameters were evaluated. Overall the treatment with HELAp, either oral or through immersion, had an anxiolytic activity similar to the treatment with buspirone. It is worth pointing out that acteoside has a modulatory effect in the Monoamine oxidase A (MAO-A) [11]. However, Carvalho [72], in a study with rats, suggested central depressor effect to this compound.

For assessment of antidepressant activity, 1% ethanol was used to induce a depressive state. The method used is adapted from the open field test for rodents by Bencan et al. [63], and was described by Blaser and Gerlai [73]; although this method is used to asses anxiety-like behavior, other studies have linked it to depression [4,20,63,74], corroborating the results from other methods [7,8,9].

The groups treated (orally or through immersion) with fluoxetine or HELAp had decreased exploratory behavior toward the lowest zones of the tank induced by 1% ethanol, chronic stress, or social isolation (*p* < 0.01; Figure 10); evidencing the antidepressant activity of HELAp and the standard antidepressant fluoxetine.

According to Barcellos et al. [75] and Abreu et al. [76], it is possible to assess the anxiety using latency as a parameter, since higher latency to enter the superior part of the tank indicates higher anxiety and higher depression. Regarding this parameter, the group treated with 1% ethanol had higher latency to explore the top, followed by the groups of chronic stress and social isolation. The groups treated with HELAp and fluoxetine had a significantly reduced period of latency to reach the top of the tanks (*p* < 0.01; Figure 11).

The number of quadrants crossed in the groups treated with 1% ethanol was lower when compared to the other groups as a consequence of decreased locomotor activity caused by the depressive effect of this substance at this concentration. In contrast, the groups treated with fluoxetine or HELAp had high values of this parameter (*p* < 0.01; Figure 12).

The erratic swim is a behavior caused by an increased reaction to fear [73,77,78,79,80]. This parameter was increased in the groups treated with 1% ethanol, stressors, and social isolation. The groups co-treated with HELAp had significantly lower numbers of erratic swims (*p* < 0.01; Figure 13).

Regarding the period of freezing, the groups treated with 1% ethanol, stressors, and social isolation had increased values of this parameter, mostly when trying to reach the superior part of the tank. In parallel, the groups treated with fluoxetine and HELAp had decreased values (*p* < 0.01; Figure 14). It is worth noting that the treatment with 1% ethanol caused difficulty in the fishes to reach the top of the tank. Chatterjee and Gerlai [81] reported that high concentrations of ethanol for low periods could induce sedative effects, neurochemical changes, and impair their movements.

According to Giacommini et al. [82], stress induces anxiety and changes zebrafish’s behavior, which can be reversed by treatment with fluoxetine. In the novel tank diving test, it was observed that the stress caused decreased swam distance and time spent at the top of the tank. Treatment with fluoxetine or HELAp managed to prevent this (*p* < 0.01; Figure 15).

Different areas of the brain are involved in different defense strategies depending on the threat level. Animal models show that in potentially dangerous situations, the structures involved are the septo-hippocampal and amygdala [83].

Caffeine is a highly fat-soluble molecule and hence easily passes through the blood–brain barrier, reaching peak concentration a few minutes after administration [84]. In the brain, caffeine blocks adenosinergic receptors (A1) and activates the enzyme nitric oxide synthase (NOS), increasing the production of nitric oxide (NO) potentializing the anxiety behavior in animals [85].

Even though previous studies have reported anxiolytic and antidepressant activity in the HELAp, our study corroborates this further in a different model and with more parameters assessed. For this, two treatment routes were tested and compared in zebrafish. The results corroborate the use of HELAp as a potential novel treatment for mood disorders. In addition, the oral treatment (the most used in humans) in zebrafish in this study makes this model even more efficient in the attempt to discover new drugs, thus allowing to compare with the treatment by immersion, which has already been consolidated by several authors who have shown that, comparing the two routes, HELAp exerted more evident pharmacological effects when administered orally in zebrafish, proving to be an effective and safe therapeutic alternative in the management of mood disorders.

## 4. Materials and Methods

### 4.1. Plant Material

The plant (*A. polystachya*) was cultivated in the garden of medicinal plants (“Farmácia da Natureza”) from Jardinópolis—SP, Brazil. The specimen was identified by Dra Lúcia Rossi from the Herbarium of the Botanical Institute of São Paulo. Therefore, the specimen was stored in the Herbarium of Medicinal Plants from the University of Ribeirão Preto—SP, Brazil, under n° HPM-1213. The permission to evaluate the biological activities of the plant extract was given by the Brazilian Institute of Environment and Renewable Natural Resources (n° 02001.005074/2011‒19).

The leaves (100 g) were dried over 72 h in a stove at 45 °C. Then, the extract was prepared through maceration in water:ethanol (20:80; *v*/*v*) over seven days and filtered through filter paper (Whatman n° 41, Sigma-Aldrich, São Paulo, Brazil). The filtered extract was slowly concentrated under low pressure at 40 °C in a rotary evaporator and freeze-dried. The process produced 12 g of dry crude extract, corresponding to a proportion of 7.1:1 between plant material:dry extract. The extract (HELAp) was stored in an amber bottle in a desiccator until used.

### 4.2. HELAp UPLC-MS Analysis

Chromatographic analyses were performed using a Waters Acquity UPLC H-Class system equipped with a diode array detector (DAD) and a Waters Xevo TQ-S tandem quadrupole mass spectrometer with a Z-spray source operating in negative ion mode. Stock solutions containing 1.0 mg/mL of the extract or the standard acteoside in LC-grade methanol were prepared separately with sonication over 30 min each and filtered through 0.45 μm Millipore filters (Merck Millipore). The solutions were diluted to 10 μg/mL with methanol, then, aliquots (5 μL) were injected onto a Sigma-Aldrich Ascentis Express C18 column (100 × 4.6 mm i.d.; 2.7 μm particle size). The mobile phase consisted of water containing 0.1% formic acid (solvent A) and methanol containing 0.1% formic acid (solvent B) supplied at a flow rate of 0.5 mL/ min according to the elution profile: isocratic with 3% B between 0 and 4 min, followed by linear gradients from 3 to 60% B between 4 and 19 min and from 60% to 90% B between 19 and 23 min, and finally returned to 3% B between 23 and 28 min. The eluent was monitored by DAD in the range of 210 to 720 nm and by MS with the optimized source and operating parameters as follows: capillary voltage 2.50 kV, Z-spray source temperature 150 °C, desolvation temperature (N2) 350 °C, desolvation gas ow 600 L/h, and mass range of m/z 150 to 600 in full-scan mode.

### 4.3. Acteoside Quantification Through HPLC-DAD

A sample (1 mg) of the dry hydroethanolic extract was redissolved in 1 mL of a mixture (80:20; *v*/*v*) of methanol (J.T. Baker HPLC grade) and Milli-Q Ultrapure water (Merck Millipore, São Paulo, Brazil), sonicated for 30 min, and filtered through a 0.45-μm Millipore filter. Aliquots (20 μL) of this solution were analyzed on a Shimadzu LC-10APvp system coupled to an SPD-M10Avp DAD and fitted with a Phenomenex Luna C18 column (250 × 4.6 mm i.d., 5 μm) protected by a Phenomenex C18 precolumn (4.0 × 3.0 mm i.d., 5 μm). Separations were carried out at room temperature (22 ± 1 °C) using a mobile phase comprising acetic acid 0.1% in water (solvent A) and methanol (solvent B; J.T. Baker HPLC grade) supplied at a constant flow rate of 1.0 mL/ min according to the program: linear gradient from 10% to 70% B between 0 and 32 min, from 70% to 10% B between 32 and 35 min, and a final isocratic elution with 10% B between 35 and 40 min. The detection wavelength was set at 330 nm.

The content of acteoside in the extract was estimated using acteoside (Sigma-Aldrich, São Paulo, Brazil; CAS no. 61276-17-3) as the external standard. Solutions containing 500, 250, 125, 62.5, 31.2, and 15.6 μg/mL of the reference standard were prepared, and calibration curves were constructed by subjecting each solution to HPLC analysis in triplicates. Analytical data were validated with respect to linearity, precision, and accuracy according to the guidelines issued by the [86].

### 4.4. Animals

In this study were used female fishes (*Danio rerio*, AB wild-type), between 6 and 8 months old, 3.5–4.0 cm length and 700–750 mg weight (216 animals in total), bought from the company Acqua New Aquarius and Fish Co. ME, Igarassu—PE. The fishes were kept in the zebrafish room with standardized water (ISO 1996), controlled temperature (26 ± 2 °C), pH (6.0 – 8.0), and light–dark cycle (12/12 h) [30,31,34,74]. The fish were fed twice a day, with brine shrimp at 9 AM and ration at 16 PM.

The protocols in this study are in accordance with the legislation from the National Council of Animals Experimentation (CONCEA). The initial project was approved by the Ethics Committee of Animals Use from the Federal University of Amapá—CEUA/UNIFAP, under n° 0002/2017.

### 4.5. Anxiety Evaluation in Zebrafish

#### 4.5.1. Drugs, Reagents, and Treatments

To evaluate the anxiolytic activity, the oral administration (2 µL/animal) was performed according to Sampaio et al. [74]. The groups (n = 12/group) were divided as follows:

Group I—Negative control, treated with distilled water;

Group II—Positive control, treated with an anxiogenic agent (caffeine at 100 mg/kg, anhydrous, Jilin Shulan Co., lote CA201609033, China);

Group III—Group treated with an anxiogenic + anxiolytic agent (caffeine at 100 mg/kg + buspirone 25 mg/kg, Libbs Farmacêutica Ltd.a., Lote 16I0258, São Paulo), the caffeine was administered 30 min after buspirone;

Group IV—Experimental group treated with anxiogenic agent + HELAp (caffeine at 100 mg/kg + HELAp at 10 mg/kg), the caffeine was administered 30 min after HELAp.

Sixty minutes after treatments, the animals were subjected to the assays, as described by Sampaio et al. [74]. The doses used in this study were chosen based on previous studies: caffeine [60], buspirone [26,63], HELAp [7,8].

As for the immersion treatments (n = 12/group), the compounds were diluted in the following concentrations: caffeine 100 mg/L, buspirone 25 mg/L, HELAp 10 mg/L; the negative control was immersed only in the system water. Each group was exposed to its respective compound over 60 min. Then, each fish was individually subjected to the scototaxis test [74].

#### 4.5.2. Scototaxis Test (Light–Dark Test)

The test was based on the method described by Maximino et al. [53,61]. For this was used a rectangular water tank measuring 15 × 10 × 45 cm (height, width, length) divided into two compartments, a white (light) and a black (dark). In the center of the tank, there was a central area delimited by mobile hatches. The water column was kept at 10 cm in height, totalling 4.5 L. A 75 W lamp was placed 1.8 m above the water tanks for lighting, and the experiments were recorded using a video camera (Nikon D5100) placed 56 cm above the test apparatus. The individual behavior analysis was performed through the software Any-maze^®^ (Stoelting CO, EUA).

After the treatments (oral and immersion), each animal was individually placed in the central area for three minutes for acclimation. After this period, the hatches were open, and the fish was recorded over 15 min. Data from animals that did not cross the center line over the 15 min were discarded to avoid false positive results of preference [87].

In the individual behavioral assessment, the following parameters were appraised: period in the white compartment (s); alternations (n, the number of times the animal crossed the central area); latency (s, the period to enter the white compartment); period of freezing (s, the period the fish kept motionless in the white compartment); erratic swimming (n, number of occurrences the fish swam in zigzag motion, with fast change of direction); thigmotaxis period (s, the proportion of time that the fish, in the white compartment, swims about two centimeters from the tank wall); risk assessment (n).

### 4.6. Antidepressant Evaluation in Zebrafish

#### 4.6.1. Drugs, Reagents, and Treatments

In the antidepressant evaluation, the animals (n = 12/group) were orally treated via gavage (2 µL/animal); 1% ethanol was used to induce depression (Solven Solvents and Chemicals Co., lote 536/16, São Paulo), fluoxetine at 20 mg/kg was used as standard antidepressant (Eurofarma Laboratories Co., lote 426954, São Paulo), HELAp was used at 10 mg/kg; the negative control group was treated with distilled water, and the animals were subjected to assay 60 min after treatment.

In the immersion treatment (n = 12/group), the compounds were diluted as follows: ethanol at 1%, fluoxetine at 20 mg/L, HELAp at 10 mg/L, and the negative control was exposed only to system water. Each group was exposed to the treatment over 30 min and then subjected to the tests.

#### 4.6.2. Developmental Social Isolation (DSI)

To ensure the social isolation of embryos during their development, they were individually separated right after fertilization. These embryos were placed in 1 L water tanks without visual, tactile or olfactory contact with other water tanks. In the control group were used five embryos per tank to allow social interaction, and their tank was also kept without visual, tactile, or olfactory contact with other tanks. The system water circulation was interrupted for eight days to prevent olfactory contact with other tanks; hence, the water was manually changed. The embryos were monitored until adults to perform the tests [74].

#### 4.6.3. Unpredictable Chronic Mild Stress (UCMS)

The stressors (six) were randomly applied, and each animal was subjected to them twice a day over 15 days [74].

Each stressor was chosen according to the previous stressor and the UCMS rules; the stressors were performed as follows:

(A) The water level from the tanks is lowered until exposure of the fish’s back over two minutes [4,25,74];

(B) The animals are transferred to another tank six times consecutively [4,25,74];

(C) The animals are persecuted with fishnet over eight minutes [4,25,74];

(D) The animals are suspended in the air using a net over two minutes [74,88];

(E) Three-quarters of the tank water is changed three times consecutively with the fishes still inside [4,25,74];

(F) The animals are individually isolated over 60 min in Eppendorf tubes (2 mL) with holes in both extremities to allow the water flux [4,25,74];

Except for the tank change stressor, all stressors were performed in the housing tank. The fishes subjected to social condition were separated only during the stressors of net and restriction and then returned to their group. All animals of treated groups received the same number and type of treatment [4,25,74].

#### 4.6.4. Novel Tank Diving Test (NTDT)

This experiment was adapted from [21,77,78,89,90]. The water tank used measured 15 × 25 × 20 cm (width, length, height), equally divided into two horizontal sections with 9.3 cm height (top and bottom); these sections were further divided resulting in eight rectangles (60.45 cm^2^) externally marked. The water column was kept at 18.6 cm in height (6.9 L).

After treatment with the depressor agent, the animals were treated with an antidepressant. After 60 min of treatment with the antidepressant agent (fluoxetine), the animals were individually transferred to the test tank, without acclimatization. Animals behavior was registered over six minutes [21]. The experiments were recorded with a video camera (Nikon D5100), placed 35 cm away from the tank, recording all the front part of the apparatus [74]. The behavioral analysis was performed through the software ANY-maze^®^ (Stoelting CO, EUA).

The following parameters were registered: period in the top (s); latency (s, the period the animal takes to reach the top); quadrants crossed (n); erratic swimming (n, number of occurrences the fish swam in zigzag motion, with fast change of direction); period of freezing (s, the period the fish kept motionless); and swam distance (m) [21].

### 4.7. Statistical Analysis

The statistical analysis was performed using the software GraphPad Prism (v 6.0). The comparison between groups was performed using one-way ANOVA followed by the post-hoc Tukey–Kramer test. The results were given as a mean ± standard deviation of the mean, and values with *p* < 0.01 were considered statistically significant.

## 5. Conclusions

The treatment with HELAp (oral and through immersion) managed to reverse the anxiogenic activity of caffeine in the fishes, without impairing their locomotion. Additionally, the treatment managed to cause antidepressant activity similar to fluoxetine; this effect is most probably due to its main compound, acteoside.

## Figures and Tables

**Figure 1 pharmaceuticals-12-00106-f001:**
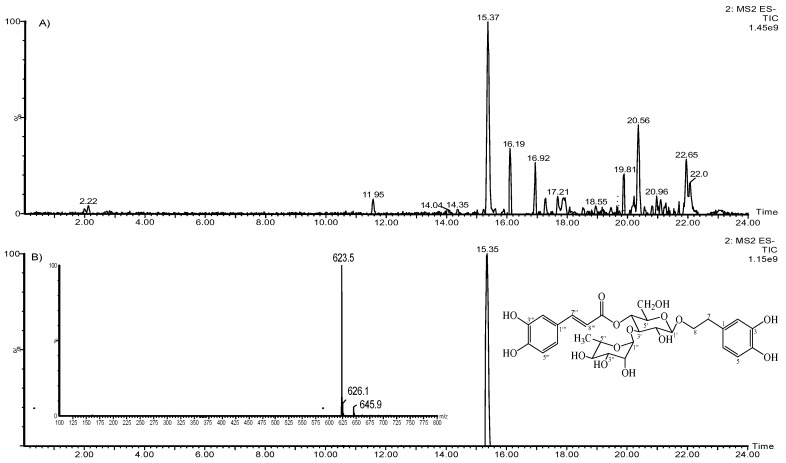
*Aloysia polystachya* hydroethanolic extract UPLC-MS chromatograms (**A**); acteoside negative ion electrospray ionization mass spectra (ESI/MS) (**B**).

**Figure 2 pharmaceuticals-12-00106-f002:**
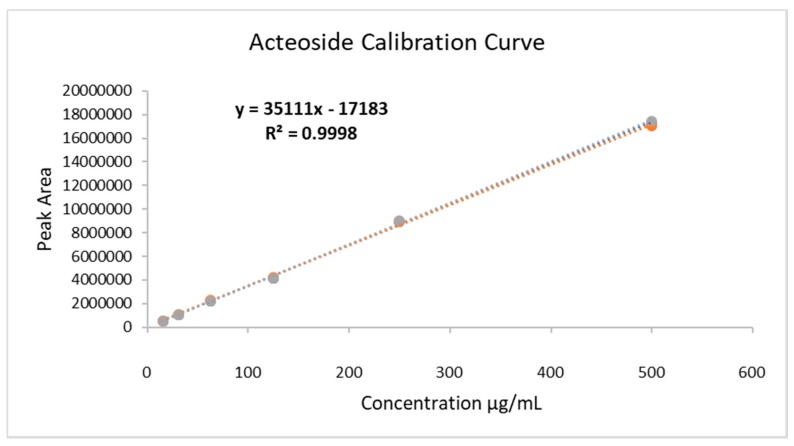
Calibration curve of standard acteoside for quantification of the analyte in hydroethanolic extracts of *A. polystachya* leaves. Concentration range: 15.25 to 500 µg/mL (R^2^ = 0.9995; R = 0.9998); LoD = 0.27 µg/mL; LoQ = 0.89 µg/mL.

**Figure 3 pharmaceuticals-12-00106-f003:**
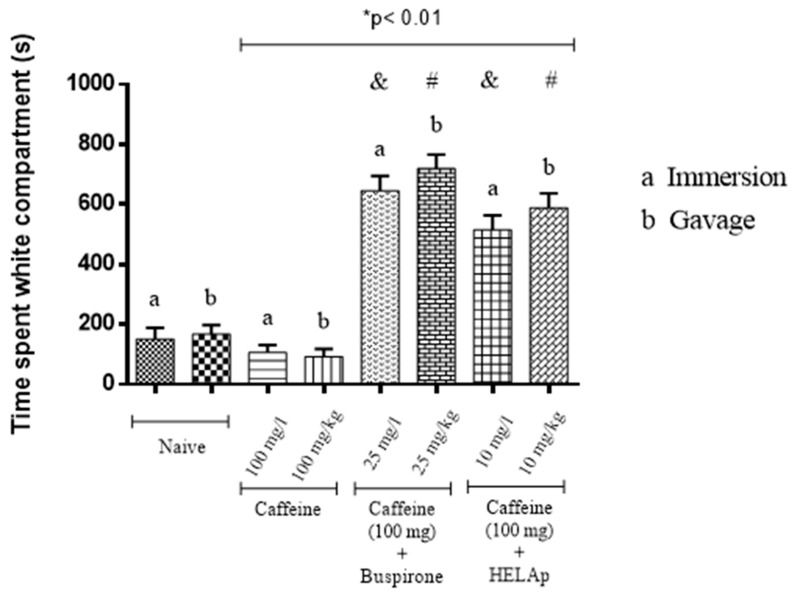
Effects of the hydroethanolic extract from *A. polystachya*, buspirone, and caffeine on zebrafish behavior in the scototaxis test (time spent in the white compartment) using immersion and oral (gavage) administration. Data are presented as a mean ± SEM (n = 4/groups); * *p* < 0.01 vs. naive fishes; & *p* < 0.01 vs. caffeine immersion; # *p* < 0.01 vs. oral caffeine administration. Statistical analysis was performed through one-way ANOVA followed by the post-hoc Tukey test.

**Figure 4 pharmaceuticals-12-00106-f004:**
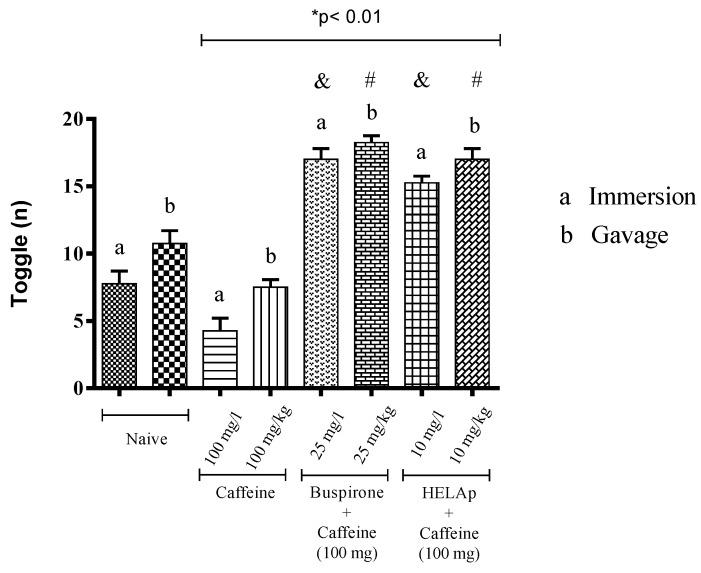
Effects of the hydroethanolic extract from *A. polystachya*, buspirone, and caffeine on zebrafish behavior in the scototaxis test (toggle) using immersion and oral (gavage) administration. Data are presented as mean ± SEM (n = 4/groups); * *p* < 0.01 vs. naive fishes; & *p* < 0.01 vs. caffeine immersion; # *p* < 0.01 vs. oral caffeine administration. Statistical analysis was performed through one-way ANOVA followed by the post-hoc Tukey test.

**Figure 5 pharmaceuticals-12-00106-f005:**
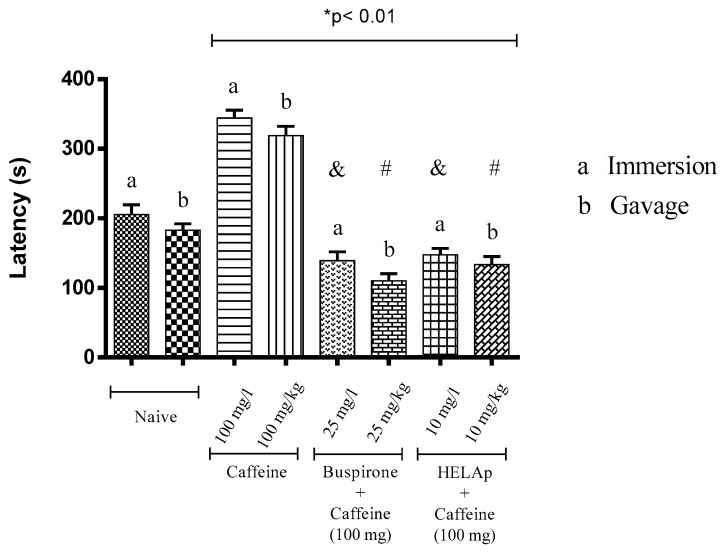
Effects of the hydroethanolic extract from *A. polystachya*, buspirone, and caffeine on zebrafish behavior in the scototaxis test (latency) using immersion and oral (gavage) administration. Data are presented as mean ± SEM (n = 4/groups); * *p* < 0.01 vs. naive fishes; & *p* < 0.01 vs. caffeine immersion; # *p* < 0.01 vs. oral caffeine administration. Statistical analysis was performed through one-way ANOVA followed by the post-hoc Tukey test.

**Figure 6 pharmaceuticals-12-00106-f006:**
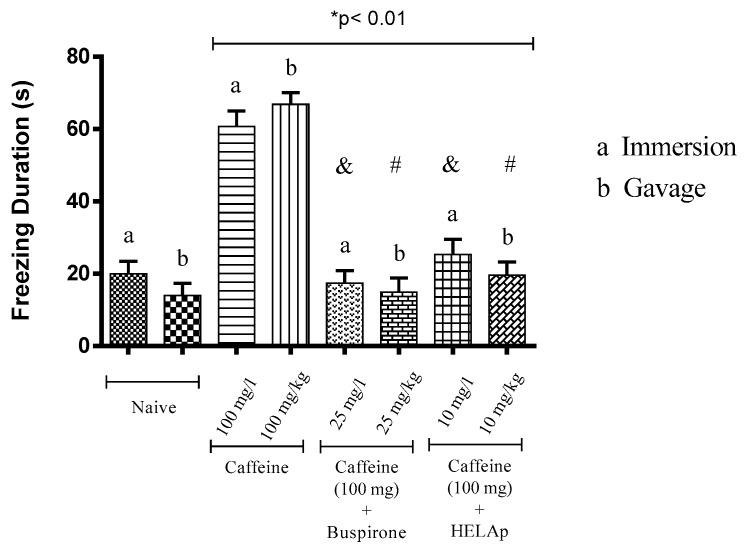
Effects of the hydroethanolic extract from *A. polystachya*, buspirone, and caffeine on zebrafish behavior in the scototaxis test (freezing duration) using immersion and oral (gavage) administration. Data are presented as mean ± SEM (n = 4/groups); * *p* < 0.01 vs. naive fishes; & *p* < 0.01 vs. caffeine immersion; # *p* < 0.01 vs. oral caffeine administration. Statistical analysis was performed through one-way ANOVA followed by the post-hoc Tukey test.

**Figure 7 pharmaceuticals-12-00106-f007:**
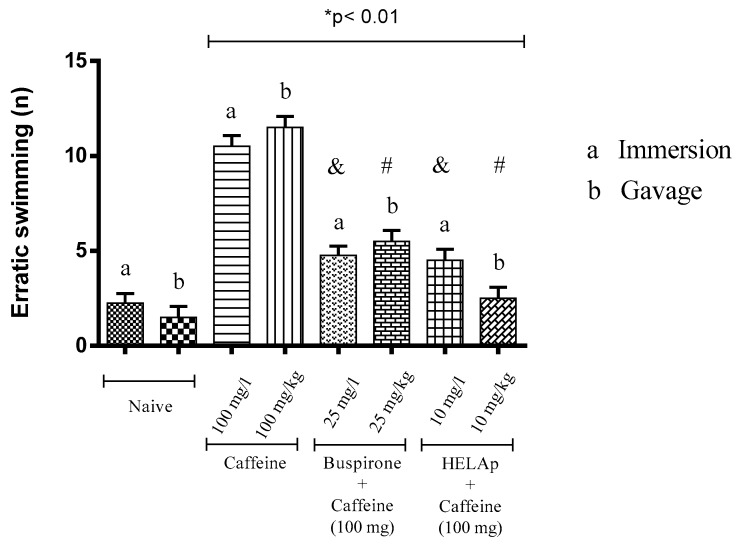
Effects of the hydroethanolic extract from *A. polystachya*, buspirone, and caffeine on zebrafish behavior in the scototaxis test (erratic swimming) using immersion and oral (gavage) administration. Data are presented as mean ± SEM (n = 4/groups); * *p* < 0.01 vs. naive fishes; & *p* < 0.01 vs. caffeine immersion; # *p* < 0.01 vs. oral caffeine administration. Statistical analysis was performed through one-way ANOVA followed by the post-hoc Tukey test.

**Figure 8 pharmaceuticals-12-00106-f008:**
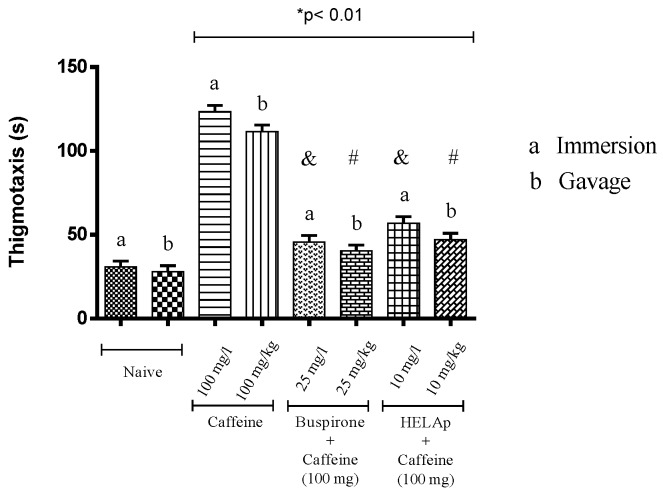
Effects of the hydroethanolic extract from *A. polystachya*, buspirone, and caffeine on zebrafish behavior in the scototaxis test (thigmotaxis), using immersion and oral (gavage) administration. Data are presented as mean ± SEM (n = 4/groups); * *p* < 0.01 vs. naive fishes; & *p* < 0.01 vs. caffeine immersion; # *p* < 0.01 vs. oral caffeine administration. Statistical analysis was performed through one-way ANOVA followed by the post-hoc Tukey test.

**Figure 9 pharmaceuticals-12-00106-f009:**
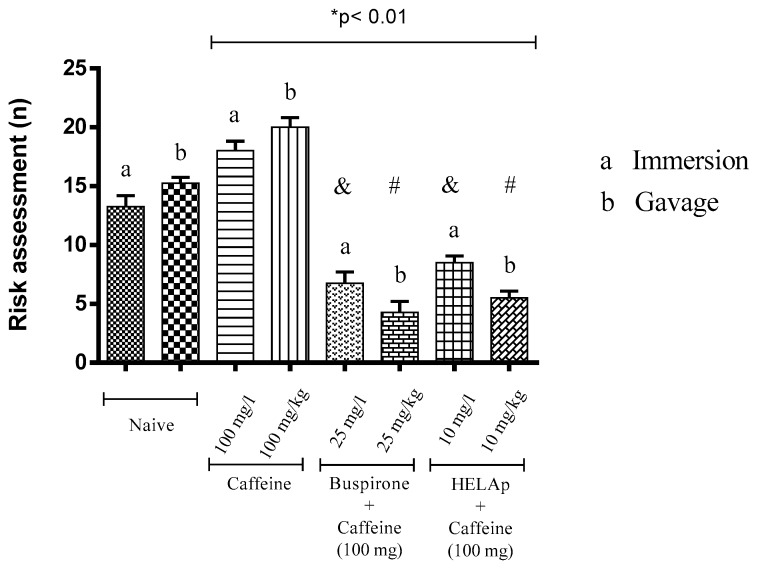
Effects of the hydroethanolic extract from *A. polystachya*, buspirone, and caffeine on zebrafish behavior in the scototaxis test (risk assessment) using immersion and oral (gavage) administration. Data are presented as mean ± SEM (n = 4/groups); * *p* < 0.01 vs. naive fishes; & *p* < 0.01 vs. caffeine immersion; # *p* < 0.01 vs. oral caffeine administration. Statistical analysis was performed through one-way ANOVA followed by the post-hoc Tukey test.

**Figure 10 pharmaceuticals-12-00106-f010:**
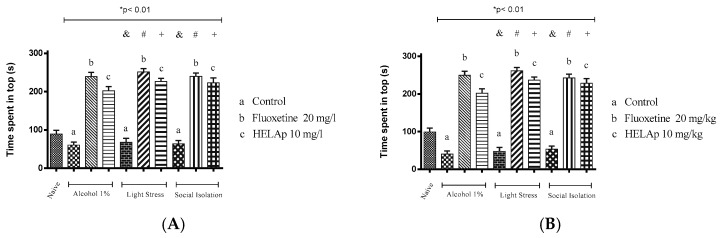
Effects of the hydroethanolic extract from *A. polystachya* on the depressive behavior of zebrafish induced through three different ways: alcohol, stressors, and social isolation in the novel tank diving test (time spent at the top of the tank). (**A**) immersion and (**B**) oral (gavage) administration. Data are presented as mean ± SEM (n = 4/group); * *p* < 0.01 vs. naïve fishes group; & *p* < 0.01 vs. alcohol control group; # *p* < 0.01 vs. light stress control group; + *p* < 0.01 vs. social isolation control group. Statistical analysis was performed through one-way ANOVA followed by the post-hoc Tukey test.

**Figure 11 pharmaceuticals-12-00106-f011:**
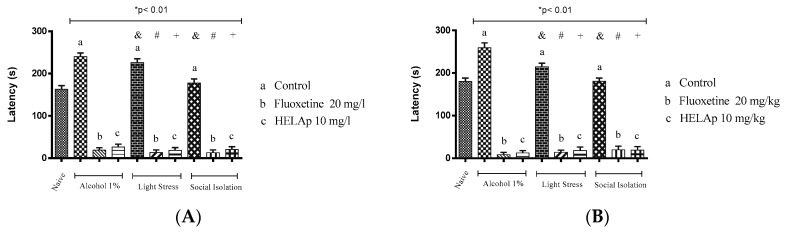
Effects of the hydroethanolic extract from *A. polystachya* on the depressive behavior of zebrafish induced through three different ways: alcohol, stressors, and social isolation in the novel tank diving test (latency). (**A**) immersion and (**B**) oral (gavage) administration. Data are presented as mean ± SEM (n = 4/group); * *p* < 0.01 vs. naïve fishes group; & *p* < 0.01 vs. alcohol control group; # *p* < 0.01 vs. light stress control group; + *p* < 0.01 vs. social isolation control group. Statistical analysis was performed through one-way ANOVA followed by the post-hoc Tukey test.

**Figure 12 pharmaceuticals-12-00106-f012:**
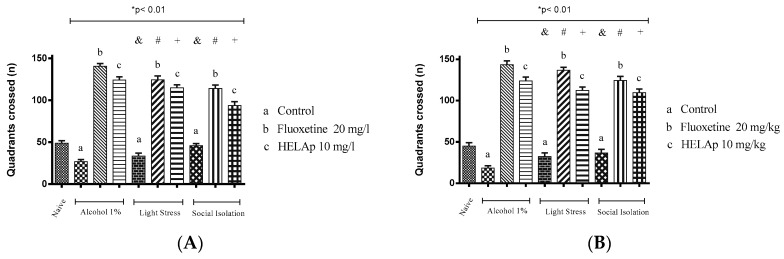
Effects of the hydroethanolic extract from *A. polystachya* on the depressive behavior of zebrafish induced through three different ways: alcohol, stressors, and social isolation in the novel tank diving test (quadrants crossed). (**A**) immersion and (**B**) oral (gavage) administration. Data are presented as mean ± SEM (n = 4/group); * *p* < 0.01 vs. naïve fishes group; & *p* < 0.01 vs. alcohol control group; # *p* < 0.01 vs. light stress control group; + *p* < 0.01 vs. social isolation control group. Statistical analysis was performed through one-way ANOVA followed by the post-hoc Tukey test.

**Figure 13 pharmaceuticals-12-00106-f013:**
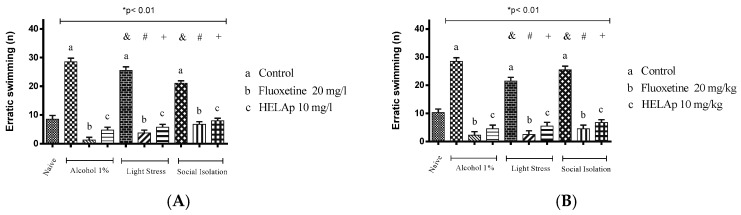
Effects of the hydroethanolic extract from *A. polystachya* on the depressive behavior of zebrafish induced through three different ways: alcohol, stressors, and social isolation in the novel tank diving test (erratic swimming). (**A**) immersion and (**B**) oral (gavage) administration. Data are presented as mean ± SEM (n = 4/group); * *p* < 0.01 vs. naïve fishes group; & *p* < 0.01 vs. alcohol control group; # *p* < 0.01 vs. light stress control group; + *p* < 0.01 vs. social isolation control group. Statistical analysis was performed through one-way ANOVA followed by the post-hoc Tukey test.

**Figure 14 pharmaceuticals-12-00106-f014:**
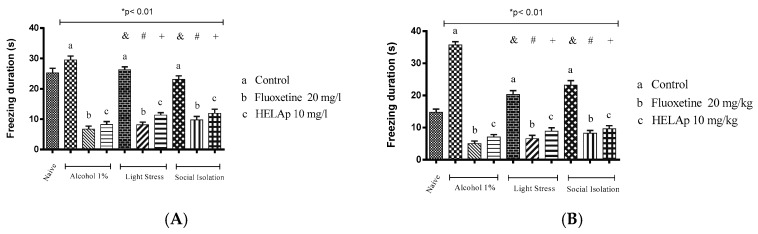
Effects of the hydroethanolic extract from *A. polystachya* on the depressive behavior of zebrafish induced through three different ways: alcohol, stressors, and social isolation in the novel tank diving test (freezing duration). (**A**) immersion and (**B**) oral (gavage) administration. Data are presented as mean ± SEM (n = 4/group); * *p* < 0.01 vs. naïve fishes group; & *p* < 0.01 vs. alcohol control group; # *p* < 0.01 vs. light stress control group; + *p* < 0.01 vs. social isolation control group. Statistical analysis was performed through one-way ANOVA followed by the post-hoc Tukey test.

**Figure 15 pharmaceuticals-12-00106-f015:**
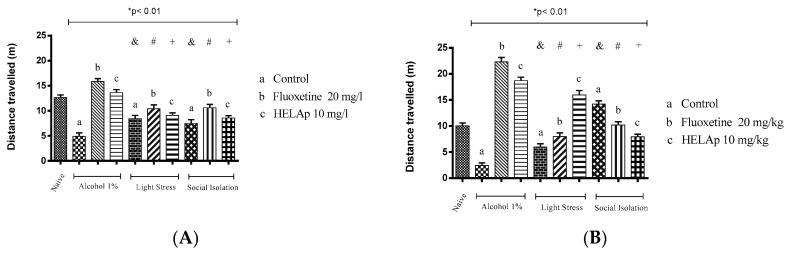
Effects of the hydroethanolic extract from *A. polystachya* on the depressive behavior of zebrafish induced through three different ways: alcohol, stressors, and social isolation in the novel tank diving test (distance travelled). (**A**) immersion and (**B**) oral (gavage) administration. Data are presented as mean ± SEM (n = 4/group); * *p* < 0.01 vs. naïve fishes group; & *p* < 0.01 vs. alcohol control group; # *p* < 0.01 vs. light stress control group; + *p* < 0.01 vs. social isolation control group. Statistical analysis was performed through one-way ANOVA followed by the post-hoc Tukey test.

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
