# Peer review of "Anxiolytic and Antidepressant Effects of the Hydroethanolic Extract from the Leaves of Aloysia polystachya (Griseb.) Moldenke: A Study on Zebrafish (Danio rerio)"

_pharmaceuticals, 2019, doi:10.3390/ph12030106_

Round 1
Reviewer 1 Report
The paper describes an anxiolytic and antidepressant effects in zebrafish of the hydroethanolic extract from the leaves of Aloysia polystachya (Griseb.) Moldenke. The main claims of the paper are properly placed in the context of the previous literature. The experimental data support the claims. An abstract represents the achievements of work. Reviewer recommends the paper to further publishing process after minor revision.
Main points:
1. The frame and construction of the introduction is poor and confused to the readers. I think the authors should reorganize introduction. What is the main purpose of study? Adaptation and validation of zebrafish procedures for evaluation of anxiolytic and antidepressant effects of the hydroethanolic extract from the leaves of Aloysia polystachya? Discovery of new drug or adjuvant for psychiatric treatment?
2. Line 52-53 - Delete
3. Some information about acteoside have to be added, for example:
Acteoside is the alpha-L-rhamnosyl-(1->3)-beta-D-glucoside of hydroxytyrosol. The hydroxy group at position 4 of the glucopyranosyl moiety has undergone esterification by formal condensation with trans-caffeic acid. Acetoside is a cinnamate ester, a disaccharide derivative, a member of catechols, a polyphenol and a glycoside. It has a role as a neuroprotective agent, an antileishmanial agent, an anti-inflammatory agent, a plant metabolite and an antibacterial agent.
4. Line 159-161 – Delete
5. Line 264-266 – it is speculation, lack of proof-Delete
6. Line 270-272 – it is speculation, lack of proof-Delete
7. Statistical analysis is required
“The ratio of peak area of standard acteoside to the corresponding concentration of analyte was established through linear regression of the standard curves. Analytical data were validated with respect to linearity, precision, and accuracy according to the guidelines issued by the [24]. Limits of detection (LoD) and quantitation (LoQ) of acteoside were 0.27 and 0.89 μg/mL, respectively.”
Author Response
The answers are attached.

Reviewer 2 Report
In the present study Costa de Melo et al. show the effects of aloysia polystachya extract (what they call HELAp) in zebrafish behaviour. In general the study seems pretty solid, and I only have some minor comments:
Introduction:
- I’d change the word “humor” for “mood” (just a matter of english style)
- I’d state that chronic stress is an animal model of depression, rather than saying that can lead to depression.
Results and Methods:
- what’s the physiological difference between given the treatment through immersion and orally. I think it would be appropriate to explain in detail this, since all the experiments are performed in both ways of delivery, and somet researchers might not be familiar with the pharmacology on zebrafish.
- The statistical analysis of choice seems a bit strange, specially since each experiment is being duplicated because of the immersion and oral treatment, more statistical power could be achieved through more factor-level analysis in the anova. But as it is I guess it is sort of enough.
- The graphs could be grouped into fewer figures, probably 2 figures would be enough: one to point out the effects in the light-dark test and another for the novel tank
Discussion:
Please reconsider talking about antidepressant-activity, in the authors own expression, the method employed is “used to assess anxiety-like behavior”, which is exactly the same that was being analyzed with the light-dark test. At least rephrase lines 231 and 232, into something like “although this method is used to asses anxiety-like behaviour, other studies have linked it to depression...”
It would be very interesting if the authors would describe the potential benefits of using these compounds as antidepressants when compared to classical antidepressants.
The study would benefit of testing the real mechanism through which HELAp works, rather than assuming it is through the modulatory effect of acteoside on MAO
Author Response
The answers are attached.
